# Classification of Drowsiness Levels Based on a Deep Spatio-Temporal Convolutional Bidirectional LSTM Network Using Electroencephalography Signals

**DOI:** 10.3390/brainsci9120348

**Published:** 2019-11-29

**Authors:** Ji-Hoon Jeong, Baek-Woon Yu, Dae-Hyeok Lee, Seong-Whan Lee

**Affiliations:** 1Department of Brain and Cognitive Engineering, Korea University, Anam-dong, Seongbuk-ku, Seoul 02841, Korea; jh_jeong@korea.ac.kr (J.-H.J.); bw_yu@korea.ac.kr (B.-W.Y.); lee_dh@korea.ac.kr (D.-H.L.); 2Department of Artificial Intelligence, Korea University, Anam-dong, Seongbuk-ku, Seoul 02841, Korea

**Keywords:** Brain-Computer Interface, electroencephalogram, mental states, drowsiness levels classification, deep learning

## Abstract

Non-invasive brain-computer interfaces (BCI) have been developed for recognizing human mental states with high accuracy and for decoding various types of mental conditions. In particular, accurately decoding a pilot’s mental state is a critical issue as more than 70% of aviation accidents are caused by human factors, such as fatigue or drowsiness. In this study, we report the classification of not only two mental states (i.e., alert and drowsy states) but also five drowsiness levels from electroencephalogram (EEG) signals. To the best of our knowledge, this approach is the first to classify drowsiness levels in detail using only EEG signals. We acquired EEG data from ten pilots in a simulated night flight environment. For accurate detection, we proposed a deep spatio-temporal convolutional bidirectional long short-term memory network (DSTCLN) model. We evaluated the classification performance using Karolinska sleepiness scale (KSS) values for two mental states and five drowsiness levels. The grand-averaged classification accuracies were 0.87 (±0.01) and 0.69 (±0.02), respectively. Hence, we demonstrated the feasibility of classifying five drowsiness levels with high accuracy using deep learning.

## 1. Introduction

A brain-computer interface (BCI) is used between human and devices for recognizing user intention. Non-invasive BCI technology allows users to communicate with external devices without brain implant surgery [1,2,3,4] using the brain’s pattern encoding [5,6] or decoding ability [7,8,9]. Over the past decades, non-invasive BCI systems [10,11] have been validated for interaction using a robotic arm [12,13,14], a wheelchair [15], and a speller [16,17,18,19]. Recently, as one of the critical issues, the BCI technology has been investigated for recognizing human mental states with high accuracy and decoding various types of mental conditions [20,21,22]. In particular, the detection of the drowsy state for autonomous driving (e.g., vehicle and aircraft) using physiological signals has been developed for artificial intelligence (AI) [23,24,25]. The drowsy state is caused by various factors such as fatigue [22,23,26,27,28], workload [22,29], and distraction [30]. Detecting the drowsy state in a driving environment has mostly been done through camera-based vision technology using human face variability. This vision technology has achieved sufficient detection accuracy [31,32]. However, if the subjects wear glasses or do not look straight ahead, the camera cannot detect drowsy state robustly [33]. Therefore, BCI-based detection for a user’s mental state is being developed because it can reflect the actual condition of the human mental states.

Some research groups have studied how to detect the drowsy state using various physiological signals, such as electrocardiogram (ECG), electrooculogram (EOG), near-infrared spectroscopy (NIRS), and electroencephalogram (EEG) signals. Fujiwara et al. [34] detected drowsy state-based multivariate statistical process control from ECG and EEG signals. A new heart rate variability (HRV)-based driver drowsiness detection algorithm is proposed by utilizing the framework of HRV-based epileptic seizure prediction. They showed 92.0% sensitivity across 34 subjects. This sensitivity value was almost the same as that obtained using conventional camera-based detection methods. Also, the grand-average false positive rate was 2.15. Hong et al. [35] also detected the drowsy state based on various machine learning methods. They measured EEG signals not only with standard 16-channel EEGs but also ear canal EEG electrodes. Also, they acquired other physiological signals such as photoplethysmography (PPG) and ECG signals. They showed the highest performance using a random forest with a kappa value of 0.98 for all subjects. In addition, the mean importance point sum of the alpha/beta power ratio was 12.76 and 11.59, respectively. Mårtensson et al. [36] decoded the drowsy state using ECG, EOG, and EEG signals. Subjective ratings using the Karolinska sleepiness scale (KSS) was used to split the data as sufficiently alert (KSS ≤ 6) or sleepy (KSS ≥ 8). KSS = 7 was excluded to get a clearer distinction between the groups. They proceeded binary classification using adaptive boosting (AdaBoost), k-nearest neighbors, linear support vector machine (SVM), Gaussian SVM, and random forest. The best performance was observed when using a random forest with 97.2% accuracy, 95.7% sensitivity, and 97.9% specificity. Awais et al. [37] recognized the drowsy state using the SVM from ECG and EEG signals in a driving environment, using a video camera to establish a ground truth for human face variability. They proposed a feature combination method of ECG and EEG signals to detect drowsy state across all subjects. They showed detection accuracies of 70.0% using ECG, 76.3%, and 80.9% using ECG and EEG signals, respectively. Nguyen et al. [38] also classified the drowsy state or not, based on Fisher’s linear discriminative analysis. EEG, EOG, ECG and NIRS signals have been measured during a simulated driving task. The blinking rate, eye closure, heart rate, alpha and beta band power were used to identify the subject’s condition. Also, a camera was installed to record the subject’s behavior. They showed 73.7% accuracy using NIRS, 70.5% accuracy using EEG, and 79.2% accuracy using NIRS+EEG across 11 subjects. Moreover, the mean of oxy-hemoglobin concentration were 0.019 and 0.017 in awake and drowsy state, respectively. In contrast, the mean of deoxy-hemoglobin concentration were −0.003 and 0.005 in awake and drowsy state, respectively.

Few research groups have investigated detecting mental states using only EEG signals. Gao et al. [23] evaluated the fatigue state based on deep neural networks using raw EEG signals. They proposed a spatio-temporal convolutional neural network (CNN) model for robust detection. They constructed deep artificial neural networks of 14 layers using a relatively small EEG dataset. In addition, by sharing the weight parameters among core blocks, the number of parameters could be reduced, which was directly related to increased performance. Wei et al. [39] detected the drowsy state from EEG signals using an SVM. They attached EEG electrodes on a non-hair-bearing area using six channels. They showed 80.0% accuracy and an area under the curve (AUC) of 0.86. Min et al. [40] compared the fatigue state as evaluated by various machine learning methods, including a neural network, random forest, k-nearest neighbors, and SVM from EEG signals. In addition, they chose significant electrodes using accuracy-based weight values. The electrodes on the left temporal-parietal region were the most significant. Dimitrakopoulos et al. [41] detected fatigue based on SVM with radial basis function from EEG signals. The subjects were asked to perform one-hour of simulated driving and 30 min of a psychomotor vigilance task in two different sessions. Once they obtained a functional brain network, they characterized the graph in terms of small-world metrics, like clustering coefficients, characteristic path lengths, and small-worldness, through graph theoretical analysis. Liang et al. [42] predicted the drowsy state based on logistic regression from EEG. They collected driving performance data and showed an AUC of 0.93 and 69% sensitivity across 16 subjects.

Accordingly, the EEG-based detection system of the drowsy state has three main advantages compared to peripheral physiological measures (PPMs) like ECG, electrodermal activity, and respiration: (*i*) accurate reflection of current mental states, (*ii*) state detection using a short length of data, (*iii*) consideration of various signal information such as spectral, temporal, and spatial aspects. However, EEG signals-based BCI system has one of the critical issues such as the constrained number of classes since non-stationary signal characteristics of the signals [43,44]. In particular, the current related studies based on EEG signals have investigated for binary classification only (i.e., alert state vs. drowsy state) due to the limited number of classes. To solve this important issue, in this paper, we focused on not only classifying the two mental states, but also on the feasibility of classifying drowsiness levels with high accuracy. We proposed a deep spatio-temporal convolutional bidirectional long short-term memory network (DSTCLN) model for classifying various drowsiness levels. In this study, drowsiness level was divided into five stages based on KSS values [45] (very alert (VA), fairly alert (FA), neither alert nor sleepy (NAS), sleepy but no effort to keep awake (SNEA), and very sleepy (VS)). We also measured EEG data related to drowsiness in the simulated night flight environment. To the best of our knowledge, this is the first attempt that demonstrates the possibility of classifying the drowsiness levels based on deep learning architecture robustly.

## 2. Methods

### 2.1. Subjects

Ten healthy subjects (S1–S10, 9 males and 1 female, aged 25.6 (±5.0)), each with over 100 h of flight experience from the Taean Flight Education Center (an affiliated educational organization of Hanseo University), participated in our experiment. All subjects were naïve BCI users and without any known neurophysiological anomalies or musculoskeletal disorders. The day before the experiment, we asked that subjects abstain from alcohol and coffee, and that they sleep for 6∼8 h the night before the experiment. Subjects were informed of the entire experimental protocol and paradigms and consented according to the Declaration of Helsinki. Experimental protocols and environment were reviewed and approved by the Institutional Review Board of Korea University [1040548-KU-IRB-18-92-A-2]. The subjects were instructed to fill out two questionnaires to check their mental and physical health conditions before the experiment to evaluate the experimental paradigm.

### 2.2. Experimental Protocols and Paradigm

The flight simulator used the Cessna 172 simulator (Garmin, Olathe, KS) as shown in Figure 1. The cockpit consisted of a monitor display, a flight yoke, and other control panels to simulate a realistic flight environment. The screen provided a 210° view of what the pilot could see outside of the aircraft. The flight yoke had a wireless KSS keypad for drowsiness level input. The keypad was attached to the flight yoke for minimizing external noise caused by movement when subjects pressed the KSS keypad.

We designed the experimental paradigm to induce the pilot’s drowsiness levels with EEG signals (Figure 2). The subjects carried out a simulated sustained night flight (high: 3000 feet, heading: 0 degrees, speed: 100 knot) for 1 h. The subjects entered KSS values for their current drowsiness level within 10 s of hearing a beeping sound. The beeping sound made the role of indexing their drowsiness level voluntarily and concentrating the experiment from inattention state due to time evolution [46]. KSS values consist of 9 drowsiness levels from KSS 1 (clearly alert state) to KSS 9 (extremely drowsy state). If the subjects failed to input KSS values or could not perform the simulated flight due to a continuous drowsy state, we regarded that interval as KSS 9.

### 2.3. Data Acquisition

The EEG and EOG signals were continuously recorded by BrainAmp devices (Brain Products GmbH, Gilching, Germany). The sampling frequency of EEG and EOG was 1000 Hz, and a 60 Hz notch filter was applied for power supply noise reduction. 30 EEG channels were placed on the scalp according to the standard international 10-20 system (Fp1-2, F3-4, Fz, FC1-2, FC5-6, T7-8, C3-4, Cz, CP1-2, CP5-6, TP9-10, P3-4, P7-8, Pz, PO9-10, O1-2, and Oz). 4 EOG channels (vertically with electrodes above and below the left eye; EOG1 and EOG2, respectively, and horizontally with electrodes at the outer canthus; left: EOG3 and right: EOG4) were also recorded. The reference and ground electrodes were placed on the FCz and AFz channels, respectively (Figure 3). During signal recording, the impedance of EEG and EOG electrodes was maintained below 10 kΩ by injecting conductive gel.

### 2.4. EEG Pre-Processing

EEG signal processing was conducted using a BBCI toolbox [47] and an OpenBMI toolbox [48] in a MATLAB 2019a environment. The EEG data were band-pass filtered between 1 to 50 Hz using a second-order zero-phase Butterworth filter and were down-sampled from 1000 Hz to 100 Hz. The data for each trial (1 min) were segmented into 1-s data without overlap as a data sample except the 10-s of data after the beep sound (Figure 2) [23]. Therefore, we could obtain 3600 samples (60 samples for a single-trial × 60 trials for an hour) for each subject. Across all subjects, for 36,000 samples in a total, we randomly selected 75% of the trials as a training set and used the remaining 25% as a test set for adopting 4-fold cross-validation. In addition, to avoid the overfitting problem for model learning, we randomly selected the data samples in a fair manner by using 75% of the samples as the training sets and the remaining 25% as the test sets for each trial [36]. We defined drowsiness levels by two approaches using the marked KSS values; (*i*) 2-class: alert state (KSS 1∼6) and drowsy state (KSS 7∼9) [36], and (*ii*) 5-class: VA (KSS 1∼2), FA (KSS 3∼4), NAS (KSS 5∼6), SNEA (KSS 7∼8), and VS (KSS 9). To obtain clear EEG data, we applied an independent component analysis (ICA) [49] to remove contaminated data components generated by eye blinks and body movement. The 4 EOG channels were used as the contaminated reference signals to reject the independent component (IC).

### 2.5. DSTCLN

We proposed the DSTCLN-based deep learning framework to classify mental states and drowsiness levels (Figure 4). In conventional studies [50,51], various EEG features such as spectral, temporal, and spatial information have been used to train deep learning models. For example, the CNN architecture could extract spatio-temporal information by considering spectral components depending on filter size [51]. Based on this characteristic, we adopted the principle of hybrid deep learning architecture (e.g., CNN+Bi-LSTM). We constructed a hierarchical CNN architecture for extracting high-level spatio-temporal features [52] and applied the Bi-LSTM network to consider time-series data characteristics like brain signals. Table 1 indicates the structure of the DSTCLN framework in detail.

The input data was composed of the total number of EEG channels and sampling points (30 × 100) after pre-processing step. The spatio-temporal CNN in the DSTCLN was designed in the form of a hierarchical CNN which was divided into five convolutional blocks to deeply extract high-level features [23]. Each convolutional block was constructed into two convolution layers and a batch normalization layer with a batch size of 32. In the convolutional blocks I, II, and III, the filter size for extracting temporal features was 1 × 5 and the stride was 1 × 1. In blocks IV and V, the filter sizes were 5 × 1 and 3 × 1, respectively, for considering spatial information. Convolutional block V was set with a 0.5 dropout ratio and the maximum-pooling and average-pooling layers were used for avoiding overfitting. We applied exponential linear units (ELUs) as activation functions in convolutional block V [51]. After filtering through the convolutional blocks, the output data sized at 1 × 256 × 76 was assigned as input data for the Bi-LSTM block.

We adopted the Bi-LSTM network to consider the long-term dependency problem for time-series data [53]. The recurrent deep learning model such as Bi-LSTM has been developed as one of the effective deep network architecture for mental state recognition based on physiological signals [54,55]. The Bi-LSTM network consists of two LSTM layers that process sequential information in two opposing directions simultaneously. One layer processes sequential instances of input data from the first-time instance to the end-time instance, and the other processes the same input data in the reverse order [53]. The final output of the Bi-LSTM at each instance in sequence is calculated by combining the two outputs of each LSTM layer. Each LSTM memory cell is processed using input data X = ($x_{1},x_{2},x_{3},\cdots ,x_{n}$) for *n* time steps as below.

At time *t*, using input $x_{t}$ and the previous hidden state $h_{t-1}$, the memory cell selects what to keep or forget from the previous states using the forget gate $f_{t}$ (1). The memory cell computes the current state $c_{t}$ in two steps. First, the cell calculates a memory cell candidate state ${\tilde{c}}_{t}$ (4). Next, using the previous cell state $c_{t-1}$ and input gate $i_{t}$ (2), the cell decides how much information to write in the current state $c_{t}$ (5). The output gate decides how much information $h_{t}$ (6) will be transferred into the next cell using the output gate $o_{t}$ (3). *W* denotes weight matrices or weight vectors, *b* denotes biases, *σ* is the logistic function and ∘ is the Hadamard product operator. (1)$$\begin{array}{cc} f_{t} & =\sigma (W_{jh}h_{t-1}+W_{fx}x_{t}+b_{f}), \end{array}$$
(2)$$\begin{array}{cc} i_{t} & =\sigma (W_{jh}h_{t-1}+W_{ix}x_{t}+b_{i}), \end{array}$$
(3)$$\begin{array}{cc} o_{t} & =\sigma (W_{oh}h_{t-1}+W_{ox}x_{t}+b_{o}), \end{array}$$
(4)$$\begin{array}{cc} {\tilde{c}}_{t} & =tanh(W_{ch}h_{t-1}+W_{cx}x_{t}+b_{c}), \end{array}$$
(5)$$\begin{array}{cc} c_{t} & =f_{t}\circ c_{t-1}+i_{t}\circ {\tilde{c}}_{t}, \end{array}$$
(6)$$\begin{array}{cc} h_{t} & =o_{t}\circ tanh\left(c_{t}\right) \end{array}$$

The proposed model consisted of four Bi-LSTM layers with 256 or 128 hidden units, and one dropout layer. The first three Bi-LSTM layers used a many-to-many approach for the input sequence of the fourth Bi-LSTM layer. The last layer used a many-to-one approach for the input into the classification block. Finally, three fully connected layers and one softmax layer were used for final decision-making in the classification block. The last fully-connected layer’s output is fed to 2 or 5-way softmax which produces a distribution over the 2 (alert or drowsy states) or 5 (five drowsiness levels) class labels [56].

We performed 50 iterations (epochs) for the model training process and saved the trained DSTCLN model weights and hyper-parameters that have the lowest loss of the training process. The final classification performance was evaluated on the test dataset in terms of classification accuracy, standard deviation (Std), sensitivity, and specificity.

## 3. Results

### 3.1. Classification Performances for Drowsiness Levels

Table 2 shows the overall classification performances of drowsiness levels across all subjects. We applied the 4-fold cross-validation method for evaluating classification accuracy fairly. The grand-averaged performance was 0.87 (±0.01) for classifying two mental states, and the sensitivity value was 0.86 and the specificity value was 0.88. In classifying five drowsiness levels, we obtained a classification performance of 0.69 (±0.02) that was a highly accurate compared to what would be expected by chance (0.2).

Figure 5 shows the confusion matrices for 2-class and 5-class classification using the DSTCLN. Each column of the confusion matrix represents the target state and each row represents the predicted state. The element (*i*, *j*) is the ratio samples in class *j* that is classified as state *i*. Figure 5a shows the results of classifying two mental states. The true positive rate for the alert state was 0.86 and the true negative rate for the drowsy state was 0.88. In contrast, the false positive and false negative rates were 0.14 and 0.12, respectively. Figure 5b indicates the results of classifying five drowsiness levels. This figure shows the highest true positive rate for the VA level at 0.88, while NAS and VS levels had the lowest values.

### 3.2. Comparison Classification Performances with Conventional Methods

Table 3 shows a comparison of classification performances using the proposed DSTCLN and conventional methods. We evaluated the classification performances for two mental states and five drowsiness levels using our experimental data. The representative conventional methods with respect to mental state detection are: power spectrum density (PSD)-SVM [57], canonical correlation analysis (CCA)-SVM [46], channel-wise CNN (CCNN) [58], EEG-based spatial-temporal CNN (ESTCNN) [23], and Deep LSTM (LSTM-D) [55].

The PSD-SVM [57] method, one of the most common conventional method, extracted spectral density power while neglecting spatial information of EEG signals, and then trained the SVM classifier. In a similar fashion, the CCA-SVM [46] method calculated the PSD corresponding to EEG channels and chose features considering the canonical correlation coefficient. After feature selection, spectral features were classified using the SVM classifier. CCNN [58], ESTCNN [23], and LSTM-D [55] methods were used to the performance comparison. The CCNN method adopted a channel-wise filter for reflecting EEG spatial adjacency to predict driver’s cognitive state. The ESTCNN [23] also considered the spatial relations and temporal dependencies using EEG signals for classifying fatigue states based on KSS values. LSTM-D [55] was comprised of a sequence-to-sequence LSTM layer and many-to-one LSTM layer for mental workload classification.

The sensitivity and specificity showed similar patterns for accuracy in classifying two mental states. The sensitivity values were 0.77, 0.73, 0.68, 0.73, and 0.55 for conventional methods, with 0.86 as the maximum sensitivity value of DSTCLN. The specificity values were 0.50, 0.84, 0.34, 0.85, 0.92 and 0.88 for each method, respectively. These results indicated that the DSTCLN model could detect the drowsy state with higher performance than the alert state across all subjects.

The comparison results showed that the proposed DSTCLN had the highest classification performance as 0.87 (±0.01) for two mental states and 0.69 (±0.02) for five drowsiness levels. The ESTCNN method achieved the highest performances with 0.78 (2-class) and 0.56 (5-class) among the conventional methods for each comparison. The proposed DSTCLN and the ESTCNN had significant performance differences for both 2-class and 5-class problems. To the statistical analysis of both two different models, we applied a paired *t*-test to the classification performance for each fold. A paired *t*-test has commonly used to determine two samples in which observations in one sample can be paired with observations in the other sample. The *p*-values as statistical results between models were observed below 0.002 and 0.001 for 2-class and 5-class problems, respectively. The CCA-SVM method showed sufficient classification accuracy in classifying two mental states but could not achieve a high performance for classifying five drowsiness levels. Although the PSD and CCA features could be extracted to robust EEG characteristics for a binary class problem, it is still limited when it comes to the multi-class problem.

### 3.3. Neurophysiological Analysis from EEG Signals

Figure 6 shows scalp topographies according to the spectral bands of EEG signals across all subjects. The scalp topographies show the grand-averaged band power to visualize brain activation during the transitional phase from the alert to drowsy state. Figure 6a,b indicate the scalp patterns corresponding to the classification of two mental states and five drowsiness levels, respectively. The amplitude was computed for all EEG channels and for each frequency band (delta, theta, alpha, beta, and gamma) across all subjects. The scalp topographies show that the amplitude was significantly different for each spectral band and brain region, not only between the two mental states in Figure 6a but also among the five drowsiness levels in Figure 6b.

As shown in Figure 6a, the amplitudes of scalp distribution between the alert state and drowsy states had significant differences in the theta, alpha, and beta bands through the paired *t*-test (*p*-value < 0.005). In contrast, in the delta and gamma bands, there is no significant difference for the amplitude variations. Additionally, the increased amplitude of theta power in the temporoparietal region, the increased amplitude of alpha power in the occipitoparietal region, and the increased amplitude of beta power in the frontal region were seen during the drowsy state compared to the alert state. In contrast, there was no difference in spatial activation between the two mental states in the delta and gamma bands.

Figure 6b for five drowsiness levels shows similar spatial patterns to the two mental states. For example, in the alpha band, the scalp distribution between the VA and VS levels showed a significant difference through the paired *t*-test (*p*-value < 0.005), but represented similar spatial patterns between each neighboring drowsiness level such as VA vs. FA and FA vs. NAS. In addition, for multiple comparisons with Bonferroni correction among the levels, we confirmed that the significant difference among VA, NAS, and VS, which are extremely discriminant drowsiness levels across all bands (*p* < 0.001). However, the *p*-values between FA and NAS for the delta, theta, alpha, and beta band were observed no significant difference as 0.56, 0.32, 0.21, and 0.11, respectively. In the case of the gamma band, the *p*-value was 0.33 between NAS and SNEA. Additionally, positive activation of theta power in the temporoparietal region, positive activation of alpha power in the occipitoparietal region, and positive activation of beta power in the frontal region appeared with increasing the drowsiness levels. There were no particular patterns in the delta and gamma bands when increasing the drowsiness levels.

We also investigated changes in the mean EEG frequency during alert and drowsy states. The mean frequency of EEG signals is used as an indicator of the general slowing of brain activity [59]. For calculating the mean frequency for all channels, we used the PSD on each spectral interval. Table 4 represented the mean frequency of each band across all subjects. This multivariate analysis revealed a significant decrease in power at high frequency and an increase at a lower frequency for all channels. In our experiment, the mean frequency of alert state was slightly higher than that of drowsy state except in theta band (difference: −0.23 Hz). In addition, we compared the mean EEG frequency for five drowsiness levels. In the case of 5-class problem, the mean frequencies were slightly different depending on the drowsiness level. The mean frequency of FA level showed the highest in alpha (10.01 Hz), beta (20.82 Hz), and gamma (37.13 Hz) bands. Also, in the VS level, the mean frequency was the lowest in alpha (9.69 Hz), beta (19.38 Hz), and gamma (36.00 Hz) bands. In addition, we found out the decreasing tendency when the drowsiness level was increased in the same bands.

Figure 7 shows the channel-wise spectral power in each frequency band. The highest difference value of theta power between VA and VS levels showed at the frontal region. Also, we could observe the discriminant difference between VA and VS levels of alpha power at the parietooccipital region. The relative change of beta power between VA and VS levels represented at the parietal region.

Figure 8 shows the PSD of alpha band corresponding to two mental states. The PSD of drowsy state was higher than that of alert state. Therefore, the activation degree of drowsy state was more than that of alert state in the alpha band.

## 4. Discussion

In this paper, we demonstrated the feasibility of simple classification for two mental states as well as multi-classification for detailed drowsiness levels based on deep learning techniques. The proposed DSTCLN model was based on the principle of hybrid deep learning frameworks, combining CNN and Bi-LSTM. The DSTCLN considered EEG features with respect to the human’s mental state using a spatio-temporal filter. To the best of our knowledge, this study is the first time to classify drowsiness levels using a deep learning technique with robust classification performance.

Recent conventional works with respect to the BCI-based detection of mental states have focused on accurate classification of user mental states using advanced machine learning algorithms and deep learning architecture [60,61,62,63,64,65]. Recognizing the mental conditions of drivers or pilots is a critical issue in systems using AI technology, such as autonomous vehicles and autopilot. Therefore, some groups have focused on detecting the drowsy state in a vehicle-driving environment [23,24,25,27,30,31], while a few groups performed the analysis of physiological signals for detecting the drowsy state in an aircraft environment [22,28,46]. In particular, aviation accidents do not happen more frequently than vehicle accidents; but, when they occur, they cause far worse casualties and large explosions. Statistically, more than 70% of aviation accidents can be attributed to human factors like pilot fatigue, and the drowsy state might be an important contributor to a large number of aviation accidents [66]. These aviation accidents not only have immediate effects (worse casualties and aircraft explosion) but could extend to secondary accidents such as harming the surrounding environment [67]. Therefore, in this study, we focused on data regarding drowsy states in a simulated flight experience with human pilots. Only volunteers with at least 100 h of flight experience were allowed to participate in this study. To induce the pilot’s drowsy state, we assigned the night flight simulation. Hence, our experimental data was able to indicate the pilot’s mental condition (alert state vs. drowsy state), and we showed that the DSTCLN could contribute in the real world by preventing not only large aviation accidents but also vehicle accidents as it can robustly detect a user’s mental condition.

Although direct performance comparison was impossible due to different experimental protocols and paradigm, we compared the classification performances to conventional methods using our experimental data. The DSTCLN showed the highest classification accuracies for both 2-class and 5-class classifications as 0.87 (±0.01) and 0.69 (±0.02), respectively. In addition, our proposed model had 0.86 of sensitivity and 0.88 of specificity for 2-class classification. Sensitivity means that the true data is classified as positive and specificity means that the false data is classified as negative. LSTM-D, which is one of the conventional methods, showed the lowest sensitivity and the highest specificity. However, the specificity of LSTM-D and DSTCLN could not achieve much difference. In the view of accuracy, sensitivity, and specificity, the DSTCLN had the significant ability to classify both 2-class and 5-class compared to the previous models. The proposed framework could consider the spectral and spatial features of EEG signals through CNN architecture and reflect the temporal information of time-series data using a Bi-LSTM network. In addition, we confirmed that certain spatial distributions were represented in the alert and drowsy states, respectively. We also found out that scalp topographies of VA and VS levels showed similar spatial patterns between the two mental states, and observed the characteristics of the transitional phase between FA and NAS levels. These results demonstrated that it could provide proper neurofeedback with BCI-users in a driving environment. In addition, in the gamma band, we could observe the significant difference among the various states of alert and drowsy through the one-way analysis of variance and paired *t*-test with Bonferroni correction for multiple comparisons. The *p*-values presented relatively low when compared to extremely discriminant drowsiness levels. Also, a little significant difference existed between alert state and drowsy state for the 2-class problem. In the context of EEG activity, the drowsy state showed a similar environment with inattention state [68]. Therefore, in our experiment, the gamma activity also showed a little significant difference except for specific drowsiness level comparisons such as between NAS level and SNEA level. This result could demonstrate the brain activities represented similar activity patterns compared to the conventional works for inattention/attention tasks [68] in our experiment.

Additionally, we have approached for the attempt to leave one subject out cross-validation, which is one of the representative performance evaluation methods for a subject-independent BCI paradigm for validation. In conventional works, few groups also adopted the leave one subject out cross-validation to detect the drowsy state or not using their proposed method with multimodal signals [36]. Our proposed DSTCLN using only EEG signals has also achieved high performance as 0.76 (±0.07) using leave one subject out cross-validation across ten subjects in the 2-class problem (i.e., drowsy state vs. alert state). Next, we also applied the same evaluation approach to classifying five drowsiness levels. For the 5-class problem, the classification performance was averaged 0.28 (±0.11) that is a little high chance-level accuracy (0.20). This result was an insufficient performance to implement the subject-independent BCI approach to real-world environment. The leave one subject our cross-validation approach has not adopted to the dedicated drowsiness level classification problem yet, since the standard index of drowsiness levels using KSS has much bias depending on each subject. The KSS indices have generally used as a correct label for clear mental states, but it is a subjective index for each subject. Therefore, we will consider investigating a standard index for labeling drowsy states objectively, to solve a inter-subject variability problem.

Furthermore, in the aspect of the real-world scenarios, the DSTCLN architecture had more layers than common deep learning architectures in the BCI fields, and therefore the computation cost for model training had much time. For more accurately and rapidly detect drowsiness levels for real-world scenarios, we will either modify the model configuration to a shallower model while maintaining high performance, or we will adopt a weight-parameter sharing method [23,51]. Since providing proper feedback to users and predicting mental states are critical techniques that could contribute to preventing accidents before they happen [24].

## 5. Conclusions and Future Works

In this paper, we presented the feasibility of classification for five pilot’s drowsiness levels using deep learning technique. We designed the experimental protocols to acquire EEG signals for the drowsy state in a simulated night flight environment. The proposed DSTCLN has been evaluated for its classification performance for the alert and drowsy states (2-class) and drowsiness levels (5-class) across ten subjects. The experimental results demonstrated sufficient classification performances for drowsiness levels using only EEG signals.

We will investigate advanced methods for classification performance improvement and modify the proposed DSTCLN framework. In addition, we will evaluate the architecture to classify the mental states for not only flight environment but also the vehicle-driving environment. We believe that this study could contribute to realizing fully autopiloted aircraft and autonomous vehicles in the future.

## Figures and Tables

**Figure 1 brainsci-09-00348-f001:**
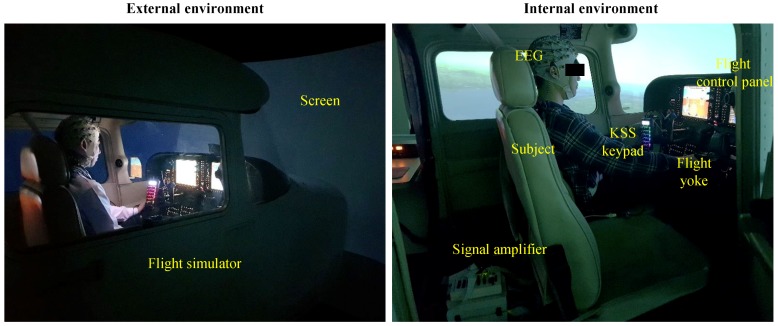
Experimental environment for EEG data acquisition in the simulated night flight.

**Figure 2 brainsci-09-00348-f002:**
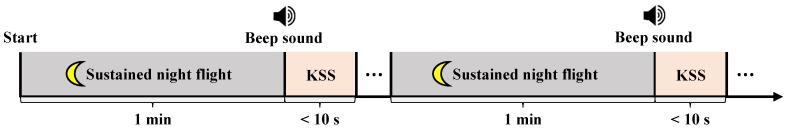
Experimental paradigm for acquiring data of various drowsiness levels.

**Figure 3 brainsci-09-00348-f003:**
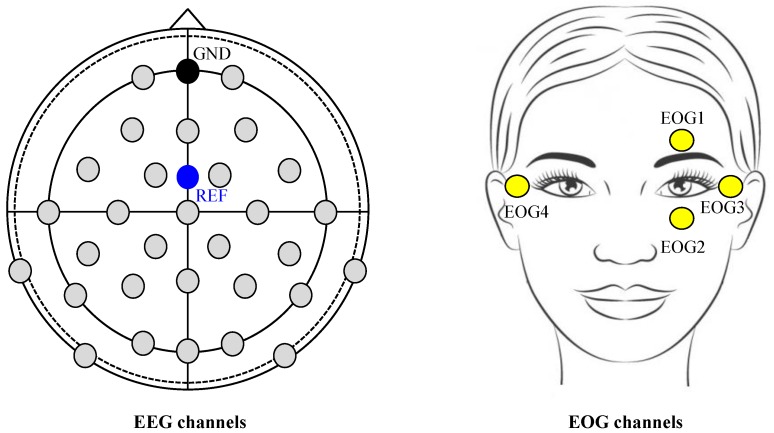
Data acquisition for 30 EEG channels and 4 EOG channels.

**Figure 4 brainsci-09-00348-f004:**
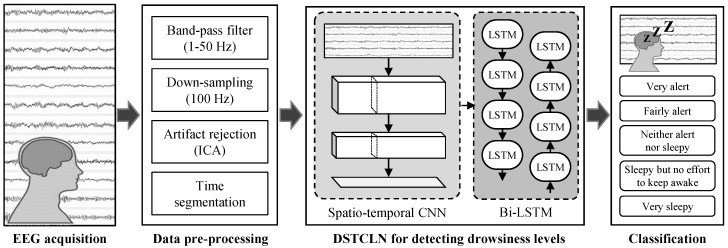
Overall flowchart for the proposed deep spatio-temporal convolutional bidirectional LSTM network (DSTCLN)-based drowsiness levels classification.

**Figure 5 brainsci-09-00348-f005:**
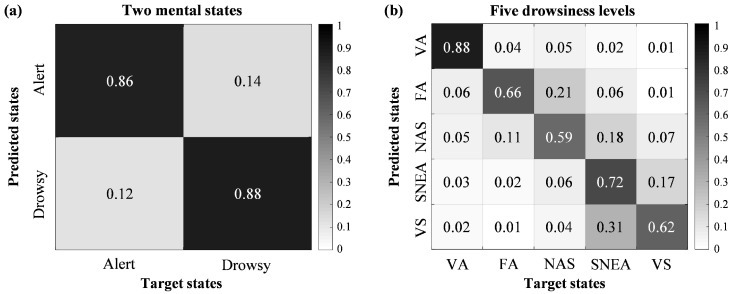
Confusion matrices of classification accuracy for each class across all subjects. (**a**) confusion matrix of classifying two mental states, (**b**) confusion matrix of classifying five drowsiness levels.

**Figure 6 brainsci-09-00348-f006:**
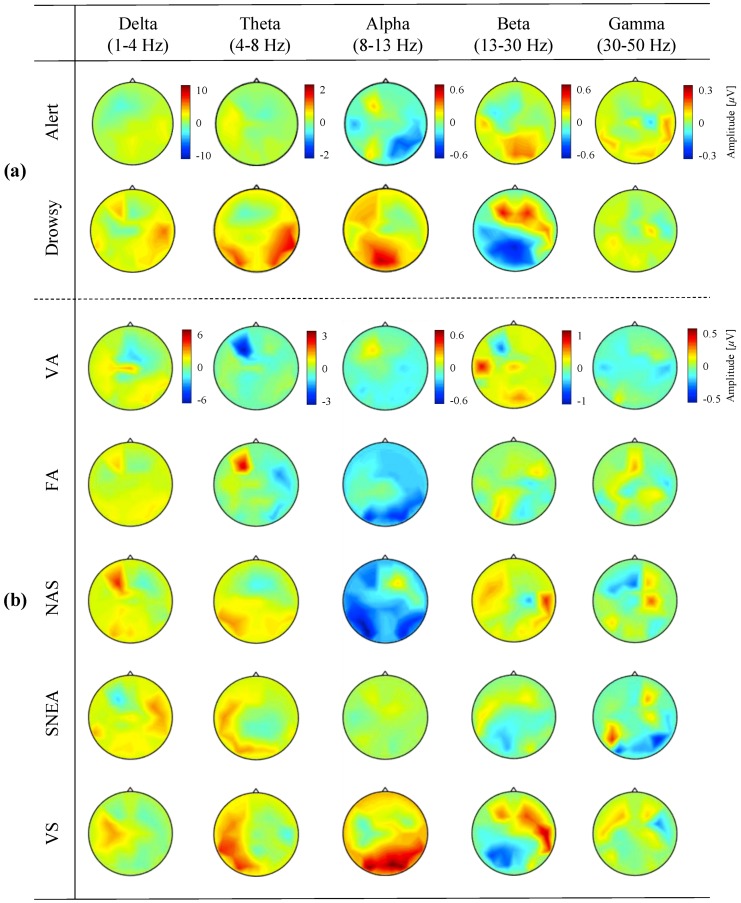
Grand-averaged scalp topographies across each spectral band (delta, theta, alpha, beta, and gamma) for two mental states (**a**) and five drowsiness levels (**b**) in all subjects. VA (very alert), FA (fairly alert), NAS (neither alert nor sleepy), SNEA (sleepy but no effort to keep awake), and VS (very sleepy).

**Figure 7 brainsci-09-00348-f007:**
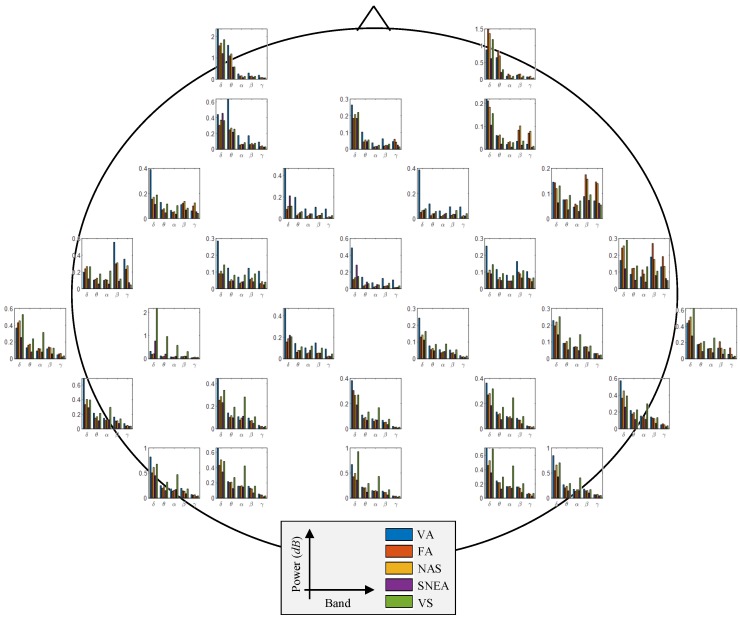
Channel-wise spectral power in each frequency band across all subjects; *δ* (delta band), *θ* (theta band), *α* (alpha band), *β* (beta band), and *γ* (gamma band).

**Figure 8 brainsci-09-00348-f008:**
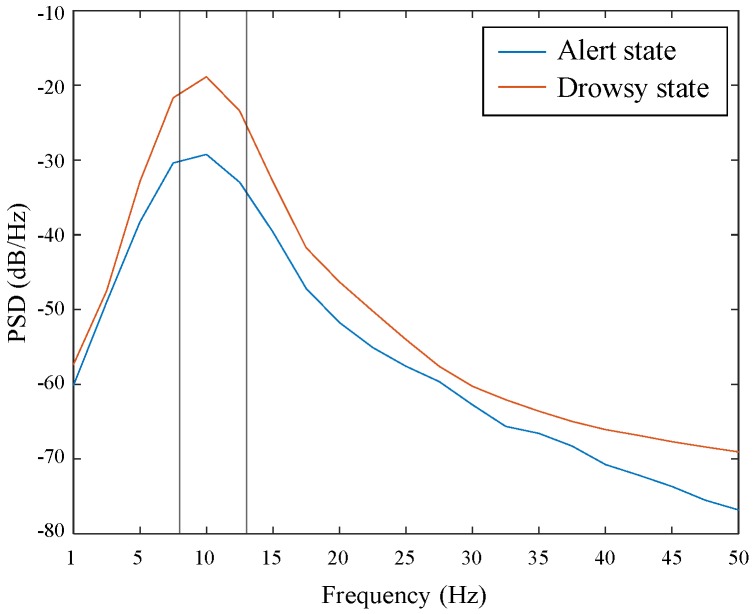
Alpha spectral peak using the PSD for alert state and drowsy state across all subjects.

**Table 1 brainsci-09-00348-t001:** The structure of the proposed DSTCLN framework.

Input Size	Block	Layer	Parameter	Output Size
-	-	Input	-	30 × 100
30 × 100	Convolutional block I	Convolution	Layer number: 2	30 × 32 × 92
			Filter size: 1 × 5	
			Feature map: 32	
			Stride size: 1 × 1	
		BatchNorm	-	
30 × 32 × 92	Convolutional block II	Convolution	Layer number: 2	30 × 64 × 84
			Filter size: 1 × 5	
			Feature map: 64	
			Stride size: 1 × 1	
		BatchNorm	-	
30 × 64 × 84	Convolutional block III	Convolution	Layer number: 2	30 × 128 × 76
			Filter size: 1 × 5	
			Feature map: 128	
			Stride size: 1 × 1	
		BatchNorm	-	
30 × 128 × 76	Convolutional block IV	Convolution	Layer number: 3	18 × 128 × 76
			Filter size: 5 × 1	
			Feature map: 128	
			Stride size: 1 × 1	
18 × 128 × 76		Maxpool	Filter size: 2 × 1	9 × 128 × 76
			Stride size: 2 × 1	
		BatchNorm	-	
9 × 128 × 76	Convolutional block V	Convolution	Layer number: 3	3 × 256 × 76
			Filter size: 1 × 3	
			Feature map: 256	
			Stride size: 1 × 1	
3 × 256 × 76		Avgpool	Filter size: 3 × 1	1 × 256 × 76
		BatchNorm	-	
		Acvivation (ELU)	-	
		Dropout (0.5)	-	
1 × 256 × 76	Bi-LSTM block	Bi-LSTM	Hidden units: 256	512 × 76
512 × 76		Bi-LSTM	Hidden units: 256	512 × 76
512 × 76		Bi-LSTM	Hidden units: 128	256 × 76
256 × 76		Bi-LSTM	Hidden units: 128	256 × 1
		Dropout (0.5)	-	
256 × 1	Classification	Fully connected	Hidden units: 128	128 × 1
128 × 1		Fully connected	Hidden units: 64	64 × 1
64 × 1		Fully connected	Hidden units: 5	5 × 1
		Softmax	-	

**Table 2 brainsci-09-00348-t002:** Overall classification performances for two mental states and five drowsiness levels.

	1-Fold	2-Fold	3-Fold	4-Fold	Classification Accuracy	Std
2-class (Alert state / Drowsy state)	0.86	0.87	0.88	0.87	0.87	±0.01
5-class (VA / FA / NAS / SNEA / VS)	0.67	0.69	0.70	0.71	0.69	±0.02

**Table 3 brainsci-09-00348-t003:** Comparison of classification performances using the conventional methods and the DSTCLN.

Methods	Alert State and Drowsy State (2-Class)				Drowsiness Levels (5-Class)	
	Accuracy	Std	Sensitivity	Specificity	Accuracy	Std
PSD-SVM [57]	0.64	0.03	0.77	0.50	0.31	0.04
CCA-SVM [46]	0.78	0.02	0.73	0.84	0.47	0.05
CCNN [58]	0.52	0.01	0.68	0.34	0.33	0.03
ESTCNN [23]	0.78	0.01	0.73	0.85	0.56	0.01
LSTM-D [55]	0.74	0.01	0.55	0.92	0.45	0.02
***Proposed*** **DSTCLN**	**0.87**	**0.01**	**0.86**	**0.88**	**0.69**	**0.02**

The text using bold type represents classification performances of the proposed DSTCLN.

**Table 4 brainsci-09-00348-t004:** EEG mean frequencies according to spectral bands in both 2-class and 5-class problem for all subjects.

	State	Delta (1–4 Hz)	Theta (4–8 Hz)	Alpha (8–13 Hz)	Beta (13–30 Hz)	Gamma (30–50 Hz)
2-class	Alert	1.99 Hz	5.98 Hz	9.86 Hz	20.74 Hz	36.96 Hz
	Drowsy	1.98 Hz	6.22 Hz	9.68 Hz	19.87 Hz	36.77 Hz
	Difference	0.01	−0.24	0.18	0.87	0.19
5-class	VA	2.19 Hz	5.82 Hz	9.92 Hz	20.68 Hz	36.59 Hz
	FA	2.22 Hz	5.92 Hz	10.01 Hz	20.82 Hz	37.12 Hz
	NAS	2.15 Hz	6.05 Hz	9.84 Hz	20.80 Hz	37.13 Hz
	SNEA	2.04 Hz	6.17 Hz	9.69 Hz	20.21 Hz	36.92 Hz
	VS	2.24 Hz	6.29 Hz	9.70 Hz	19.38 Hz	36.00 Hz

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
