# Peer review of "Classification of Drowsiness Levels Based on a Deep Spatio-Temporal Convolutional Bidirectional LSTM Network Using Electroencephalography Signals"

_brainsci, 2019, doi:10.3390/brainsci9120348_

Round 1

Reviewer 1 Report

The article " Classification of Drowsiness Levels based on a Deep Spatio-Temporal Convolutional Bidirectional LSTM Network using Electroencephalography Signals" is a interesting and relevant article in which authors found the feasibility of classifying of five drowsiness levels with high accuracy using deep learning of Electroencephalography signals.  

However, there are few issues in the article that need to be addressed as discussed below.

Introduction.

Lines 32 to 48 describe the studies that have been carried out to detect drowsiness, it is suggested to add the gold standard used to calculate the diagnostic accuracy, sensitivity and specificity of each drowsiness classification technique for more precise analysis of the articles.

Material and Methods

In line 83, “ Ten healthy subjects (S1-S10, 9 males and 1 female, aged 25.6 (55.0)),”  It would be also more clear if is the mean age and  the value of the standard deviation since it is very high (55 years ?).

In study design it would have been interesting to compare the results of a basal state with subjects awake with eyes closed, eyes open without performing any task. This would allow us to know if the changes in theta, alpha and beta power are due to alterations in attention, in working memory or to the eyes closing that accompanies drowsiness.

Results

In Figure 6, You could clarify if the power units are in µV or µV/Hz2?

It will be more clear if you could provide and analyze the power frequency spectrum of the different states to see the changes in the amplitude, in mean frequency of each band, especially of the alpha spectral peak.

Discussion

You may discuss the results obtained in the study and contrast with previous studies.  

You may also discuss why there are no significant differences in the gamma power band during the various states of alertness and drowsiness since this EEG activity is one of the most involved in the processes of attention and working memory. ¿Why the gamma activity was not modified when the attention periods decrease in the different states of drowsiness?

Reviewer 2 Report

This study presents a DSTCLN-based deep learning method for classification of mental states and drowsiness levels. The method was tested on a dataset with 10 healthy subjects and the classification results are presented for two and five classes . The manuscript is well prepared and the method is well explained. I have few concerns regarding the manuscript which are listed below.

Please clarify the number of samples for each subject in the classification process. Please clarify how the test and training sets are selected. If they are selected randomly, the neighboring samples may be selected in both train and test sets which would give the classifier access to the correct label for samples extremely similar to the test sample. I think it’s important to see how the proposed method performs in leave one subject out scenario. Please present the results for leave one subject out cross validation. No matter how the performance would be, it gives us some insight about the system.

Reviewer 3 Report

The overall paper is sound, with an adequate review of the current methods and, as far as I know, an innovative method to detect the fatigue.

However, overall, I feel that the paper is not doing enough analysing of the results for a journal in this field. Further analysis to try to provide a more in-deep understanding of the fatigue would be appreciated in a journal of brain science. Even if my recommendation is to accept the paper with minor revisions, I consider that this paper may be more apt for a journal with a different focus than brain sciences.

I have minor comments for every section:

Introduction. The authors do a good work showing the current status of the field, and provide a good literature review. However, when comparing the EEG-based system with systems based on other technologies, I think that the authors are being biased and are not presenting the problems of EEG-based system.

Materials and Methods. In general the explanation of the DSTCLN approach (an in particular the LSTM), assumes a lot of concepts that may not be known by the reader. I would suggest to either include more explanation or provide some reference with those details.

Results. As mentioned before, from the point of view of brain sciences I think there may not be enough analysis of the EEG data for a journal in this area. Also the only analysis present is not introduced in the materials, which makes it difficult to understand the reason behind this analysis. Additionally, for the same reason, there are concepts in this analysis that are not clear. To what statistical analysis do the authors refer in this section when they report p-values? In general the statistical analysis performed in this section (there are no other statistical analysis on the paper), are not explained at all.

From the point of view of the classification system, I think it would be interesting to see both the distribution and temporal evolution of the fatigue scores to see if there is any bias in the distribution, or any temporal trend. Regarding the temporal trend, did the 4-fold cross-validation used data from random points, or were the 4 folds non-overlapping blocks?

When comparing the proposed approach to the other methods, it would be interesting to see not only the accuracy but also the error (either squared or absolute) of the prediction as if this was a regression problem instead of a classification. Is not the same classify VS as FA, than VS.

Round 2

Reviewer 2 Report

The authors have answered the questions raised up by the reviewer in the last round review and the paper has improved in quality.